# Exploration of Trends in Antimicrobial Use and Their Determinants Based on Dispensing Information Collected from Pharmacies throughout Japan: A First Report

**DOI:** 10.3390/antibiotics11050682

**Published:** 2022-05-18

**Authors:** Yuichi Muraki, Masayuki Maeda, Ryo Inose, Koki Yoshimura, Naoki Onizuka, Masao Takahashi, Eiji Kawakami, Yoshiaki Shikamura, Naotaka Son, Makoto Iwashita, Manabu Suzuki, Masayuki Yokoi, Hirokazu Horikoshi, Yasuaki Aoki, Michiyo Kawana, Miwako Kamei, Hajime Hashiba, Choichiro Miyazaki

**Affiliations:** 1Department of Clinical Pharmacoepidemiology, Kyoto Pharmaceutical University, Kyoto 607-8414, Japan; inose2019@mb.kyoto-phu.ac.jp (R.I.); ky17380@ms.kyoto-phu.ac.jp (K.Y.); ky17074@ms.kyoto-phu.ac.jp (N.O.); 2Division of Infection Control Sciences, Department of Clinical Pharmacy, School of Pharmacy, Showa University, Tokyo 142-8555, Japan; m-maeda@pharm.showa-u.ac.jp; 3Takahashi Pharmacy, Tokyo 135-0005, Japan; famiyaku.t@dream.com; 4Kawakami Dispensing Pharmacy, Kyoto 617-0814, Japan; qqkr8ks9k@rondo.ocn.ne.jp; 5Faculty of Pharmaceutical Sciences, Tokyo University of Science, Tokyo 133-0033, Japan; shika@rs.tus.ac.jp; 6Pharmacy Co., Ltd., Tokyo 153-0062, Japan; n.son@pharmacy-net.co.jp; 7Totosato Pharmacy, Nagano 384-1103, Japan; i.makoto0602@gmail.com; 8Gifu Pharmaceutical Association, Gifu 500-8146, Japan; suzunoki88@outlook.jp; 9Pascal Pharmacy, Ogaki 525-0031, Japan; polestar932@nike.eonet.ne.jp; 10Marusho Pharmacy, Osaka 543-0023, Japan; hirokazu.horikoshi@gmail.com; 11Japan Pharmaceutical Association, Tokyo 160-8389, Japan; aurora@pharmacy.vip.co.jp (Y.A.); mkawana333@gmail.com (M.K.); m.kamei@thu.ac.jp (M.K.); h-hashiba@amall.co.jp (H.H.); chomiya@bronze.ocn.ne.jp (C.M.); 12Faculty of Pharmaceutical Sciences, Teikyo Heisei University, Funabashi 270-0193, Japan; 13Miyazaki Pharmacy Co., Ltd., Nagasaki 852-8116, Japan

**Keywords:** antimicrobial resistance, antimicrobial use, surveillance, defined daily dose

## Abstract

The purpose of this study was to evaluate the defined daily doses (DDD)/1000 prescriptions/month (DPM) as a new indicator that can be used in pharmacies, and to describe antimicrobial use patterns in pharmacies nationwide in Japan. Dispensing volumes, number of prescriptions received, and facility information were obtained from 2638 pharmacies that participated in a survey. DPM was calculated based on the dispensing volume and number of prescriptions, which are routinely collected data that are simple to use. Use of third-generation cephalosporins, quinolones, and macrolides in pharmacies that received prescriptions primarily from hospitals or clinics decreased from January 2019 to January 2021. In particular, the antimicrobial use was higher in otorhinolaryngology departments than in other departments, despite a decrease in the antimicrobial use. In the linear multiple regression analysis, otorhinolaryngology department was independently associated with the third-generation cephalosporin, quinolone, and macrolide prescription in all periods. This study reveals for the first-time trends in antimicrobial use through a new indicator using the volume of drugs dispensed in pharmacies throughout Japan. Antimicrobial use differed by the medical department, suggesting the need to target interventions according to the department type.

## 1. Introduction

Antimicrobial agents play an important role in healthcare and have contributed significantly to curing infections and improving patient prognosis [1]. Conversely, inappropriate antimicrobial use (AMU) has become a global problem as consumption of antimicrobials is associated with the development of antimicrobial resistance (AMR) [2]. In 2015, the World Health Organization (WHO) asked Member States to develop and enforce national action plans against AMR [3]. Against this background, Japan formulated a National Action Plan on AMR in 2016 to promote appropriate AMU [4].

In Japan, the proportions of third-generation cephalosporins, quinolones, and macrolides used are higher than in other countries [5]. Therefore, a goal was set of reducing the use of these agents by 50% by 2020 [4]. Trends in AMU in Japan have been studied using databases of sales, claims, and information on specific health checkups. Studies have shown that AMU of total antimicrobials, cephalosporins, fluoroquinolones, and macrolides has decreased over time [6,7]. However, AMU based on data from individual pharmacies and clinics is largely unknown.

AMU in Japan is mostly oral (93%) [5], and mostly prescribed in outpatient settings [8]. Japan has a universal health insurance system that provides all citizens with insurance and uniform access to medical care [9]. In Japan, pharmacies that dispense drugs and provide patient services based on the insurance system are defined as insurance pharmacies [10]. All systemic antibacterial drugs require a prescription, and citizens cannot purchase them at insurance pharmacies without a prescription. In addition, more than 70% of outpatients visiting hospitals and clinics receive prescription drugs at insurance pharmacies, with the remaining 30% receiving them at hospitals and clinics [10]. Furthermore, the percentage of patients receiving prescription drugs at insurance pharmacies is increasing in Japan as the country moves toward greater separation of labor in the pharmaceutical industry [10]. Though not all, in Japan, pharmacies that dispense prescription medication are separated from hospitals and clinics as a matter of health policy. Because patients are free to choose which pharmacy they use to collect prescription medication, we speculated the characteristics of the pharmacy might have an effect on the antibiotics dispensed. Therefore, active intervention by pharmacists in insurance pharmacies is essential to promote appropriate AMU. AMR control measures need to be implemented based on a detailed understanding of the situation in the community, and the process indicator, AMU, needs to be determined in order for the pharmacist at the dispensing pharmacy to intervene.

To evaluate the promotion of appropriate AMU by pharmacists working in insurance pharmacies, it is necessary to identify trends in AMU in each insurance pharmacy and the characteristics of AMU according to the institutional setting. AMU in outpatients in each region has been described, and AMR-reduction measures tailored to regional characteristics are being promoted in some other countries [11]. In these countries, sources of AMU are generally collected through systems that use data from medical institutions in real time [12,13,14] or by manual methods [15]. In Japan, a system has been established for collecting AMU data from hospitals [16], but there is currently no system for collecting AMU data from clinics and insurance pharmacies.

The defined daily doses (DDD)/1000 inhabitants per day (DID), which uses the amount of antimicrobial, day of therapy (DOT)/1000 inhabitants per day (DOTID), which uses the duration of administration [17], and the number of patients/1000 inhabitants per day (PID), which uses the number of patients to whom antibiotics are administered [18], have been used to assess AMU. In Japan, patients are free to choose the insurance pharmacy that they use and may use an insurance pharmacy that is outside their area of residence [19]. Therefore, it is difficult to evaluate the AMU of insurance pharmacies using DID, DOTID or PID, which use the resident population as a correction factor. In other countries, the number of prescriptions containing antimicrobial agents has been used to identify AMU in outpatients [11]. However, in Japan, the number of prescriptions containing antimicrobial agents for each insurance pharmacy is not continuously monitored. Information that can easily be collected in insurance pharmacies includes the quantity of antibiotics dispensed and the number of prescriptions received.

Hence, this study aimed to define DDDs/1000 prescriptions/month (DPM) as a new AMU indicator that could be used in insurance pharmacies and to identify AMU in insurance pharmacies in order to support the National Action Plan on AMR and to contribute to the development of indicators for continuous monitoring of AMU. Furthermore, to identify the characteristics of AMU in each pharmacy, AMU was surveyed according to the medical department from which the prescription originated, and factors influencing AMU were explored.

## 2. Materials and Methods

### 2.1. Data Sources

The survey periods were January and June of 2019 and 2021. In this study, pharmacists conducted the study and analyzed the data collected. Dispensing volumes and number of prescriptions received, and facility information were obtained from insurance pharmacies that agreed to participate in the nationwide survey. This survey was not based on a database analysis, but instead AMU and facility information calculated at each pharmacy was collected by prefectural pharmaceutical associations, and Japan Pharmaceutical Association consolidated these data. The drugs surveyed were oral antibiotics of code J01, according to Anatomical Therapeutic Chemicals developed by WHO Collaborating Centre for Drug Statistics Methodology [20].

### 2.2. Calculation of AMU

As a new indicator of AMU, DPM (DDDs/1000 prescriptions/month) was defined based on the volume dispensed at insurance pharmacies. The equation used is shown below (1). The DDD (g) values were taken from the ATC/DDD Index 2020 [21].
DPM (DDDs/1000 prescriptions/month) = DDDs/number of prescriptions received per month × 1000(1)
where DDDs = quantity of each antibacterial agent dispensed per month × potency (g)/DDD (g).

### 2.3. Statistical Analysis

The characteristics of pharmacies participating in the survey were compared using Fisher’s exact test or the Kruskal–Wallis test. Linear multiple regression analysis was then performed to identify factors associated with third-generation cephalosporin, quinolone, and macrolide DPM for each of the four study periods using pharmacy DPM data. DPM was used as the outcome variable in the regression model. Over 70% of the prescriptions received from each specific medical institution type, number of prescriptions received (per 1000 increment), number of medical institutions from which prescriptions were received (in increments of 50), facility type (clinic or hospital), and the departments from which prescriptions most frequently originated, were selected as covariates.

Multicollinearity was assessed using the Pearson correlation coefficient and VIF. All statistical analyses were two-tailed, with *p* < 0.05 indicating statistical significance. Statistical analyses were performed using SPSS statistics software version 23.0 (IBM Japan, Tokyo, Japan).

## 3. Results

### 3.1. Characteristics of Participating Pharmacies

A total of 2638 insurance pharmacies participated in this survey. The breakdown of participating pharmacies by region was as follows: Hokkaido (50), Tohoku (532), Kanto (333), Chubu (558), Kinki (369), Chugoku (169), Shikoku (142), and Kyushu/Okinawa (485). Table 1 shows the characteristics of the participating pharmacies. The number of prescriptions received, the number of medical facilities from which the prescriptions were received, and the concentration rate in each period were compared. For all measures, there were significant differences between groups (*p* < 0.001 for all). The frequency distribution of specific medical departments did not differ by year (*p* > 0.999).

### 3.2. Trends in Antimicrobial Use Based on Dispensing Information

Trends in AMU, categorized by the source of the prescriptions received by pharmacies, are shown in Figure 1 and Table A1 in Appendix A. Use of third-generation cephalosporins, quinolones, and macrolides in pharmacies that primarily accepted prescriptions from hospitals decreased by 42.5%, 27.3%, and 30.8%, respectively, from January 2019 to January 2021. Dispensing of third-generation cephalosporins, quinolones, and macrolides in pharmacies that primarily accepted prescriptions from clinics decreased 38.5%, 38.2%, and 50.1%, respectively, from January 2019 to January 2021. Other antibiotics prescribed included penicillin with extended spectrum (J01CA), β-lactamase-containing penicillin (J01CR), second-generation cephalosporins (J01DC), and sulfamethoxazole-trimethoprim (J01EE) were also dispensed in pharmacies that mainly accepted prescriptions from hospitals (Table A1 in Appendix A).

### 3.3. Trends in Antimicrobial Use According to Hospital or Clinic Characteristics

Figure 2 shows the trends in AMU classified by hospital and clinic characteristics in 2019 and 2021. Regardless of hospital or clinic characteristics, AMU in 2021 was almost always lower than in 2019 (Figure 2a). However, AMU in dermatology departments remained fairly constant and did not decrease. Otorhinolaryngology departments had the highest AMU throughout the study period. The other antibiotics prescribed in general hospitals included penicillin with extended spectrum (J01CA, Figure 2b), sulfamethoxazole-trimethoprim (J01EE), β-lactamase-containing penicillin (J01CR), and second-generation cephalosporins (J01DC). In contrast, tetracyclines (J01AA) were frequently prescribed in dermatology and obstetrics and gynecology departments, and beta-lactamase-containing penicillin (J01CR) was frequently prescribed in otorhinolaryngology departments.

### 3.4. Trends in Antimicrobial Use Based on Dispensing Information

Table 2, Table 3 and Table 4 show the results of multiple regression analysis of third-generation cephalosporin, quinolone, and macrolide DPM in 2019 and 2021. Multivariate analysis revealed that otorhinolaryngology departments were significantly associated with the use of these antibiotics. Additionally, dermatology departments were significantly associated with the use of third-generation cephalosporins and macrolides, and pediatrics departments were significantly associated with the use of third-generation cephalosporins. No strong correlation among predictors other than internal medicine departments and high variance inflation factor (VIF) values, were observed. Internal medicine departments were excluded from the final model because they were correlated with high VIF values.

## 4. Discussion

This study reveals trends in AMU in Japanese insurance pharmacies based on dispensing volumes for the first time. Third-generation cephalosporins, quinolones, and macrolides showed a downward trend in AMU from 2019 to 2021, reflecting the national trend. Therefore, the new DPM indicator based on dispensing volume could be used to assess the impact of the AMR control action plan in insurance pharmacies in Japan. In addition, problems such as departments with high AMU and low use of narrow-spectrum antimicrobials were also identified, and further contributions to the appropriate AMU by pharmacy pharmacists will be required in the future.

In Japan, a National Action Plan on AMR control was formulated in 2016 [4] and a subsequent decrease in oral AMU was reported [6]. In this study, a similar trend was observed. The newly defined DPM indicator used in this study uses dispensing volume and number of prescriptions, which is information that is routinely collected and can easily be acquired. Therefore, DPM could be used as an alternative indicator for understanding AMU by insurance pharmacies in Japan.

In the Netherlands and Belgium, a system for identifying prescribing trends in outpatient care has been implemented [12,13]. In Japan, although the Ministry of Health, Labour and Welfare consolidates information on insured medical care, it is strictly regulated and is not readily accessible. In addition, there is no system to aggregate the characteristics of each pharmacy. Therefore, this study collected DPM and facility information aggregated individually from each pharmacy. In the future, it is necessary to establish a system to collect information in real time in Japan and to utilize it for policymaking. However, this methodology can be calculated by using only the number of antimicrobials dispensed and total number of prescriptions filled by a pharmacy, so it should be easy to use regardless of whether the country is a developed or developing country.

In this study, the number of prescriptions decreased over the study period. A decrease in the number of health service visits as a result of COVID-19, in addition to AMR-prevention measures, could have contributed to the decrease in AMU from 2019 to 2021. Although we speculated that there would be seasonal effects in the prescription of antibiotics, widespread COVID-19 in 2021 had a greater impact than season on prescription trends.

While some countries show similar trends to Japan [13,22], in other countries the opposite trend has been observed because of telemedicine consultation resulting in overcautious physicians prescribing more antibiotics when faced with the limited diagnostics available [12,23]. Prescriptions of antibiotics increased in hospitals in some countries during the COVID-19 pandemic because physicians tended to prescribe “prophylactic antibiotics” to hospitalized COVID-19 patients almost by default [14,24,25]. It seems that Japan may be among the few countries where these general patterns were not observed. The COVID-19 pandemic is ongoing and ongoing monitoring is needed to assess its impact on antibiotic prescriptions.

Conversely, the number of medical facilities that provided prescriptions increased, and the proportion of prescriptions received per medical facility decreased. Since 2015, family pharmacists in Japan have been required to collaborate with prescribers and medical institutions to monitor patients’ medication status centrally and continuously [26]. Against this backdrop, the number of outpatients in Japan who received their medications at insurance pharmacies has increased over time among those who visit hospitals and clinics [10]. Thus, a series of healthcare reforms is likely to have contributed to this trend.

The types of antimicrobials and AMU differ by region [11], number of beds in medical facilities [27], age [8,11,28,29], sex [28,29], number of drugs prescribed [29] and department [30]. In this study, the main determinant of AMU identified was the department from which the prescriptions originated. Specifically, the use of antimicrobials was higher in otorhinolaryngology departments than in other departments, despite a decrease in the use of antimicrobials. As in internal medicine, otorhinolaryngology has many patients with upper respiratory tract infections as well as chronic sinusitis and otitis media [31]. The DPM calculated in this study uses the number of prescriptions received as a correction factor, suggesting that the frequency of antimicrobial prescriptions, the amount used, and the number of days of administration are high as a percentage of prescriptions received. Macrolides are sometimes used for anti-inflammatory purposes in chronic sinusitis [32] and quinolones are effective against otitis media [33], and they are recommended as an option for moderate and severe chronic otitis media in Japanese treatment guidelines [34], suggesting that they may be used universally. In the future, the purpose of AMU in otorhinolaryngology should be investigated to evaluate the appropriateness of their use.

Dermatology was dominated by tetracycline use. This may be because tetracyclines are effective in acne vulgaris and pemphigoid [35,36]. In multivariate analysis, dermatology was significantly associated with the prescription of third-generation cephalosporins and macrolides. Because infectious diseases typically treated in dermatology, such as impetigo and cellulitis, are caused by *Staphylococcus* spp. and *Streptococcus* spp. there may be overuse of broad-spectrum antibiotics [37]. Sulfamethoxazole and trimethoprim have been approved for use in general hospitals. Sulfamethoxazole-trimethoprim is recommended for the prevention of *Pneumocystis* pneumonia [38]. Because general hospitals include a variety of departments, antimicrobial agents may have been prescribed for a wide range of infections. Beta-lactamase-containing penicillins and second-generation cephalosporins were also widely used. Japanese hospitals have antimicrobial stewardship teams that promote the appropriate use of antimicrobials [39,40]. Therefore, in general hospitals, the use of narrow-spectrum (targeted) antimicrobials may be promoted by the intervention of the antimicrobial stewardship team. However, since the trend is not observed elsewhere, further support for appropriate use by pharmacy pharmacists is needed in outpatient clinics and departments.

Our finding that otorhinolaryngology was independently associated with third-generation cephalosporin, quinolone, and macrolide use in all periods reinforces the importance of health policy [41,42,43]. In 2018, a revised medical reimbursement system was implemented, with the introduction of an antimicrobial stewardship (AS) fee in pediatric clinics to reduce unnecessary antibiotic prescriptions [44,45,46]. However, the system did not include pediatric patients seen in otorhinolaryngology departments. This study suggests that the AS fee should be applied to otorhinolaryngology patients with sinusitis or otitis media, which are conditions for which broad-spectrum antibiotics may be overused. The Ministry of Health, Labour and Welfare revised the reimbursement standard in 2022 and extended it to otorhinolaryngology patients. Ongoing nationwide surveillance is required to monitor the effects of the change in policy.

The study has several limitations. First, AMU was calculated based on dispensed quantities at insurance pharmacies, making it difficult to assess the purpose of use and the patient background. Second, the DPM measure may underestimate AMU in children and patients with impaired renal function because they require lower maintenance doses. Despite these limitations, the trends in AMU in insurance pharmacies identified in this study provide important information that could be used to further promote appropriate AMU. Further evaluation of the DPM measure through interventions by pharmacy pharmacists and comparison with the situation in other countries is warranted.

## 5. Conclusions

This study revealed, for the first time, trends in AMU through a new indicator using the volume of drugs dispensed in insurance pharmacies throughout Japan. Therefore, we propose that this methodology could be used as an indicator to support national action plans on AMR and to monitor AMU in pharmacies on an ongoing basis. This DPM measure can be calculated using only the number of antimicrobials dispensed and the total number of prescriptions filled in pharmacies, so in theory, it should be easy to use regardless of whether the country is a developed or developing country. AMU differed depending on the main source departments from which the prescriptions originated, suggesting the need to implement appropriate measures specific to each department. In Japan, it is necessary to promote appropriate use, especially in otolaryngology and dermatology.

## Figures and Tables

**Figure 1 antibiotics-11-00682-f001:**
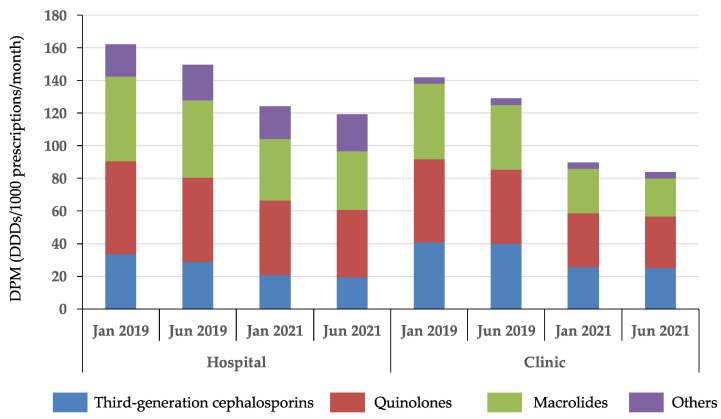
Trends in antimicrobial use based on dispensing information collected from various pharmacies in 2019 and 2021. The four bars on the left show antimicrobial use in pharmacies where the prescriptions received are mainly from hospitals. The fours bars on the right show antimicrobial use in pharmacies where the prescriptions received are mainly from clinics. Values represent the median DPM (defined daily doses/1000 prescriptions/month).

**Figure 2 antibiotics-11-00682-f002:**
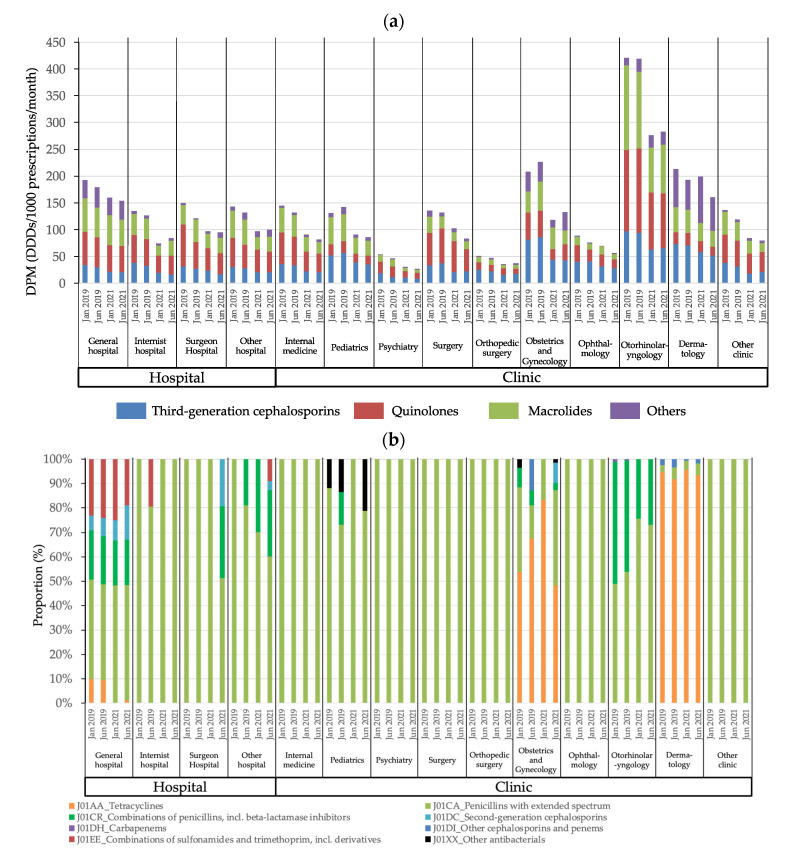
Trends in antimicrobial use classified by hospital or clinic characteristics based on dispensing information collected from various pharmacies in 2019 and 2021. (**a**) Trends in antimicrobial use classified by hospital or clinic characteristics; (**b**) Breakdown in antimicrobial use other than quinolones, third-generation cephalosporins, and macrolides classified by hospital or clinic type. Values represent the median DPM (daily defined doses/1000 prescriptions/month).

**Table 1 antibiotics-11-00682-t001:** Characteristics of the pharmacies that participated in the survey.

	January 2019	June 2019	January 2021	June 2021	*p*
Number of monthly prescriptions received per pharmacy per month *	1324.5	1206.5	1018.0	1152.0	<0.001
	(880.0, 1902.0)	(812.0, 1790.8)	(689.3, 1515.0)	(780.0, 1710.0)	
Number of medical facilities/departments from which the prescriptions were received, per pharmacy per month *	39	40	39	42	<0.001
	(25, 61)	(25, 63)	(26, 63)	(28, 67)	
The concentration rate, % *	85.0	83.6	81.3	81.7	<0.001
	(61.1, 93.4)	(59.2, 92.7)	(57.0, 91.7)	(56.2, 92.0)	
Source of the prescription, n					>0.999
General hospital	432	437	450	453	
Internist hospital	90	90	92	91	
Surgeon hospital	20	20	20	19	
Other hospital	55	58	57	57	
Internal medicine	992	979	1016	1011	
Pediatrics	153	151	143	152	
Psychiatry	83	86	95	90	
Surgery	31	32	32	32	
Orthopedic surgery	123	131	136	134	
Obstetrics and gynecology	10	11	9	10	
Ophthalmology	86	91	92	94	
Otorhinolaryngology	168	169	169	169	
Dermatology	149	158	167	170	
Other clinic	125	130	133	138	

* Values represent the median (interquartile range) (defined daily doses/1000 prescriptions/month).

**Table 2 antibiotics-11-00682-t002:** Linear regression model for predictions of third-generation cephalosporins prescription.

Factors	January 2019 (Winter)				June 2019 (Summer)				January 2021 (Winter)				June 2021 (Summer)			
	β	95%CI		*p*	β	95%CI		*p*	β	95%CI		*p*	β	95%CI		*p*
Constant term	12.6	−117.5	142.7	0.849	27.6	−65.7	121.0	0.562	31.8	−1.7	65.3	0.063	26.3	−21.3	74.0	0.279
Over 70% of prescriptions received from a specific medical institution	15.6	−24.4	55.5	0.445	3.9	−25.0	32.7	0.792	4.0	−6.2	14.1	0.445	−1.1	−15.6	13.5	0.887
Number of prescriptions received (increments of 1000)	−0.1	−17.6	17.4	0.991	−2.2	−15.2	10.8	0.739	1.4	−4.0	6.8	0.617	2.2	−4.9	9.2	0.544
Number of medical institutions from which prescriptions were received (increments of 50)	−2.1	−20.3	16.1	0.821	−0.5	−14.8	13.9	0.951	1.5	−3.8	6.8	0.582	−0.9	−8.0	6.2	0.805
Pharmacy that prescriptions were received mainly from a clinic	23.2	−39.9	86.2	0.472	15.7	−29.7	61.1	0.497	1.3	−15.1	17.6	0.881	4.5	−18.8	27.8	0.704
Most common source of the prescriptions																
General hospital	−1.0	−75.7	73.8	0.980	−7.5	−61.4	46.4	0.785	−12.2	−31.6	7.2	0.217	−6.4	−34.1	21.3	0.650
Internal medicine hospital	−1.0	−97.3	95.4	0.984	−3.9	−73.8	65.9	0.912	−2.4	−27.3	22.5	0.850	1.6	−34.3	37.5	0.931
Surgery hospital	−15.4	−213.5	182.7	0.879	−20.1	−163.5	123.3	0.783	−6.9	−58.8	44.9	0.793	−4.9	−80.9	71.1	0.899
Other hospitals	5.5	−126.6	137.7	0.934	2.1	−91.5	95.7	0.965	−4.6	−38.7	29.5	0.791	−0.2	−48.8	48.3	0.992
Pediatrics	149.2	73.2	225.2	<0.001	126.4	70.8	182.0	<0.001	60.8	40.3	81.2	<0.001	76.9	48.5	105.4	<0.001
Psychiatry	−33.4	−135.7	68.9	0.522	−31.0	−103.7	41.8	0.404	−21.3	−46.3	3.8	0.096	−18.8	−55.6	17.9	0.315
Surgery	−26.3	−186.3	133.7	0.747	−7.2	−121.2	106.8	0.901	1.8	−39.4	43.0	0.933	13.0	−46.0	72.1	0.665
Orthopedic surgery	−25.7	−109.3	57.9	0.546	−24.6	−83.5	34.3	0.413	−13.4	−34.3	7.5	0.209	−11.9	−42.0	18.3	0.440
Obstetrics and gynecology	70.2	−205.2	345.6	0.617	71.8	−118.6	262.2	0.460	61.9	−13.9	137.6	0.109	121.1	17.7	224.5	0.022
Ophthalmology	−10.9	−108.3	86.6	0.827	−9.3	−78.4	59.7	0.791	4.2	−20.6	28.9	0.741	6.8	−28.4	42.0	0.706
Otorhinolaryngology	126.5	53.0	199.9	0.001	105.4	52.2	158.6	<0.001	63.6	44.5	82.7	<0.001	83.8	56.4	111.2	<0.001
Dermatology	28.2	−49.3	105.7	0.476	68.2	12.7	123.7	0.016	33.9	14.5	53.3	0.001	34.4	6.5	62.3	0.016
Other clinics	9.3	−73.6	92.2	0.826	4.7	−54.4	63.8	0.875	8.8	−12.3	29.9	0.413	19.1	−10.7	48.8	0.209

The partial regression coefficient indicates a variation of DPM when each factor was present. β, partial regression coefficient; CI, confidence interval. Background color indicates *p* < 0.05.

**Table 3 antibiotics-11-00682-t003:** Linear regression model for predictions of quinolones prescription.

Factors	January 2019 (Winter)				June 2019 (Summer)				January 2021 (Winter)				June 2021 (Summer)			
	β	95%CI		*p*	β	95%CI		*p*	β	95%CI		*p*	β	95%CI		*p*
Constant term	198.8	−125.6	523.2	0.230	158.5	−7.7	324.6	0.062	100.7	39.9	161.6	0.001	110.2	−79.2	299.7	0.254
Over 70% of prescriptions received from a specific medical institution	67.0	−32.6	166.7	0.187	17.5	−33.8	68.8	0.503	21.0	2.5	39.5	0.026	−27.5	−85.3	30.3	0.351
Number of prescriptions received (increments of 1000)	−0.2	−43.7	43.4	0.993	9.1	−14.0	32.2	0.439	11.8	2.0	21.7	0.018	12.9	−15.2	40.9	0.369
Number of medical institutions from which prescriptions were received (increments of 50)	−8.6	−53.9	36.7	0.710	−5.2	−30.8	20.4	0.691	0.7	−9.0	10.3	0.892	−16.9	−45.1	11.3	0.239
Pharmacy that prescriptions were received mainly from a clinic	−42.7	−200.0	114.6	0.594	−23.6	−104.4	57.1	0.566	−26.2	−55.9	3.6	0.085	−15.0	−107.8	77.7	0.751
Most common source of the prescriptions																
General hospital	−108.4	−294.8	77.9	0.254	−84.8	−180.8	11.2	0.083	−34.1	−69.3	1.1	0.058	−11.9	−122.0	98.1	0.832
Internal medicine hospital	−64.9	−305.2	175.4	0.596	−34.6	−158.9	89.7	0.585	−15.0	−60.3	30.2	0.515	−14.7	−157.3	128.0	0.840
Surgery hospital	−893.3	−577.3	410.7	0.741	−78.9	−334.0	176.2	0.544	−26.9	−121.0	67.2	0.575	−22.2	−324.3	279.9	0.885
Other hospitals	−135.6	−465.1	193.9	0.420	−92.8	−259.3	73.8	0.275	−36.3	−98.2	25.5	0.249	−25.3	−218.3	167.8	0.798
Pediatrics	−101.5	−291.0	88.0	0.294	−85.0	−183.9	13.9	0.092	−33.1	−70.1	4.0	0.080	−29.0	−142.2	84.2	0.616
Psychiatry	−123.2	−378.3	131.9	0.344	−100.5	−230.0	28.9	0.128	−53.9	−99.3	−8.5	0.020	−46.2	−192.3	100.0	0.536
Surgery	−93.9	−492.9	305.1	0.644	−54.9	−257.7	147.9	0.596	−13.5	−88.3	61.3	0.723	−20.8	−255.5	213.8	0.862
Orthopedic surgery	−117.8	−326.3	90.8	0.268	−101.7	−206.5	3.1	0.057	−49.8	−87.7	−11.9	0.010	−41.8	−161.5	77.9	0.494
Obstetrics and gynecology	−78.8	−765.6	608.0	0.822	−65.5	−404.2	273.3	0.705	−45.1	−182.6	92.5	0.520	−3.0	−414.0	408.0	0.989
Ophthalmology	−94.4	−337.3	148.5	0.446	−89.2	−212.0	33.7	0.155	−32.5	−77.5	12.4	0.156	−37.9	−177.8	102.0	0.595
Otorhinolaryngology	286.1	103.0	469.2	0.002	147.3	52.7	241.9	0.002	112.5	77.8	147.1	<0.001	293.8	185.0	402.6	<0.001
Dermatology	−97.4	−290.7	95.8	0.323	−76.7	−175.4	22.1	0.128	−36.7	−72.0	−1.4	0.042	−40.5	−151.6	70.5	0.474
Other clinics	−50.4	−257.1	156.2	0.632	−31.4	−136.6	73.8	0.558	−4.5	−42.7	33.8	0.820	12.6	−105.6	130.7	0.835

The partial regression coefficient indicates a variation of DPM when each factor was present. β, partial regression coefficient; CI, confidence interval. Background color indicates *p* < 0.05.

**Table 4 antibiotics-11-00682-t004:** Linear regression model for predictions of macrolides prescription.

Factors	January 2019 (Winter)				June 2019 (Summer)				January 2021 (Winter)				June 2021 (Summer)			
	β	95%CI		*p*	β	95%CI		*p*	β	95%CI		*p*	β	95%CI		*p*
Constant term	103.1	−93.0	299.3	0.302	120.6	21.4	219.8	0.017	90.9	38.3	143.5	0.001	76.4	27.7	125.1	0.002
Over 70% of prescriptions received from a specific medical institution	27.1	−33.1	87.4	0.377	6.3	−24.3	37.0	0.685	3.5	−12.5	19.5	0.668	1.6	−13.3	16.4	0.838
Number of prescriptions received (increments of 1000)	4.1	−22.2	30.4	0.760	6.3	−7.5	20.1	0.372	9.0	0.5	17.5	0.039	5.6	−1.6	12.8	0.126
Number of medical institutions from which prescriptions were received (increments of 50)	−8.0	−35.4	19.4	0.565	−6.2	−21.5	9.1	0.426	−3.9	−12.3	4.4	0.356	−4.3	−11.6	2.9	0.240
Pharmacy that prescriptions were received mainly from a clinic	2.2	−92.9	97.3	0.964	−15.9	−64.2	32.3	0.517	−19.6	−45.3	6.1	0.135	−17.2	−41.0	6.7	0.159
Most common source of the prescriptions																
General hospital	20.0	−92.7	132.6	0.728	−2.6	−59.9	54.7	0.929	33.1	2.6	63.6	0.033	43.7	15.4	72.0	0.002
Internal medicine hospital	−11.8	−157.0	133.5	0.874	0.1	−74.1	74.3	0.997	6.3	−32.8	45.4	0.753	25.4	−11.3	62.1	0.175
Surgery hospital	−66.4	−365.0	232.3	0.663	−46.7	−199.0	105.7	0.548	−27.9	−109.3	53.4	0.501	−15.0	−92.7	62.6	0.704
Other hospitals	−61.6	−260.8	137.6	0.544	−52.8	−152.2	46.7	0.298	−28.2	−81.6	25.3	0.302	−14.1	−63.7	35.6	0.578
Pediatrics	−47.3	−161.9	67.3	0.418	−19.0	−78.1	40.1	0.529	−1.1	−33.1	31.0	0.948	1.8	−27.3	30.9	0.904
Psychiatry	−77.9	−232.1	76.3	0.322	−56.2	−133.5	21.1	0.154	−29.7	−69.0	9.5	0.138	−20.3	−57.9	17.3	0.290
Surgery	−69.7	−310.9	171.4	0.571	−56.7	−177.8	64.4	0.359	−33.3	−97.9	31.4	0.313	−22.2	−82.6	38.1	0.470
Orthopedic surgery	−89.0	−215.1	37.0	0.166	−74.2	−136.8	−11.6	0.020	−40.3	−73.0	−7.5	0.016	−28.8	−59.5	2.0	0.067
Obstetrics and gynecology	−11.7	−426.8	403.5	0.956	−12.8	−215.0	189.5	0.902	−6.7	−125.6	112.2	0.912	5.7	−100.0	111.4	0.916
Ophthalmology	−74.1	−220.9	72.7	0.322	−61.9	−135.3	11.5	0.098	−20.8	−59.6	18.1	0.294	−19.1	−55.1	16.8	0.297
Otorhinolaryngology	183.5	72.9	294.2	0.001	177.1	120.6	233.6	<0.001	119.8	89.8	149.7	<0.001	153.4	125.4	181.4	<0.001
Dermatology	−7.5	−124.3	109.4	0.900	12.1	−46.9	71.1	0.687	30.8	0.3	61.3	0.048	28.7	0.1	57.2	0.049
Other clinics	−47.5	−172.4	77.4	0.456	−35.6	−98.4	27.2	0.266	−6.3	−39.4	26.8	0.707	3.0	−27.4	33.4	0.846

The partial regression coefficient indicates a variation of DPM when each factor was present. β, partial regression coefficient; CI, confidence interval. Background color indicates *p* < 0.05.

## Data Availability

The data presented in this study are available on request from the corresponding author.

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
