# Peer review of "Exploration of Trends in Antimicrobial Use and Their Determinants Based on Dispensing Information Collected from Pharmacies throughout Japan: A First Report"

_antibiotics, 2022, doi:10.3390/antibiotics11050682_

Round 1
Reviewer 1 Report
The work presents some interesting results in a clear and direct way for define DDDs/1,000 prescriptions/month (DPM) as a new AMU indicator that can be used in insurance pharmacies, and to describe AMU in insurance pharmacies nationwide. Furthermore, to identify the characteristics of AMU in each pharmacy, AMU was surveyed according to the medical department from which the prescription originated, and factors influencing AMU were 86
explored.
- The manuscript contains a series of small errors that do not detract from the merit of the work but do diminish its quality and presentation and it is necessary to correct them before being accepted. The introduction is direct and very short. A little more should be said about the daily doses (DDD)/1,000 prescriptions/month (DPM) as a new indicator that can be used in pharmacies.
- Very little 11 References in introduction.
- Also 28 References little for reserach article, at least from 50 References.
- The font in tables 2-4 very small, you can make page horizontal and incresre the font.
- Resolution of figures need to imrovment.
- The survey periods were January and June of 2019 and 2021 and now we are nereast june 2022. Why you are late in publication the data? it is necessary the data updated.
-
A criterion must be unified to express numerical data. Correct the entire manuscript. Review the format of bibliographic references so that it is unified.
Best regards,
Author Response
Reviewer 1 comments:
The work presents some interesting results in a clear and direct way for define DDDs/1,000 prescriptions/month (DPM) as a new AMU indicator that can be used in insurance pharmacies, and to describe AMU in insurance pharmacies nationwide. Furthermore, to identify the characteristics of AMU in each pharmacy, AMU was surveyed according to the medical department from which the prescription originated, and factors influencing AMU were 86 explored.
The manuscript contains a series of small errors that do not detract from the merit of the work but do diminish its quality and presentation and it is necessary to correct them before being accepted.
Response
We would like to thank you for reviewing our manuscript and for your helpful suggestions. The changes to the manuscript are shown in red font. We have provided a point-by-point response to your comments below.
The introduction is direct and very short.
Response
We have expanded the introduction based on your suggestions and those of the other reviewers. (Page 2, Lines 59-80 and 85-89; Page 3, Lines 101-106)
A little more should be said about the daily doses (DDD)/1,000 prescriptions/month (DPM) as a new indicator that can be used in pharmacies.
Response
According to your suggestion, we have expanded on this in the introduction (Page 2, Lines 85-89, and Page 3, Lines 101-106).
Very little 11 References in introduction.
Response
According to your suggestion, we have revised the introduction and added several references. There are 19 references cited in the revised discussion. (Page 2, Lines 59-80 and 85-89; Page 3, Lines 101-106)
Also 28 References little for research article, at least from 50 References.
Response
According to your suggestion, we have added several references. The revised manuscript has a total of 46 references. According to the instructions to authors articles are required to have at least 30 references.
The font in tables 2-4 very small, you can make page horizontal and increase the font.
Resolution of figures need to improvement.
Response
We have been modified the format of Tables 2-4 and increased the size of the figures, based your comments.
The survey periods were January and June of 2019 and 2021 and now we are nearest June 2022. Why you are late in publication the data? it is necessary the data updated.
Response
This survey was not a database analysis, but rather AMU and facility information calculated at each pharmacy was collected by prefectural pharmacetical associations, and the Japan Pharmaceutical Association consolidated these data. Data collection and processing are time-consuming. Currently there is no ongoing collection of data in real time. This is something to work toward in the future. Publication of our manuscript would help us to obtain the necessary support and buy-in. Based on your comments, we have added some wording to the Methods (Page 3, Lines 115-118) and Discussion (Page 10, Lines 225-235) sections.
A criterion must be unified to express numerical data. Correct the entire manuscript. Review the format of bibliographic references so that it is unified.
Response
Explanations of numerical data are included in the footnotes to the figures and tables. The format of the references has also been checked by an English proofreader.
Reviewer 2 Report
The study is interesting for the content and the results are clearly presented. Minor spell check is required in the methods section.
Author Response
Reviewer 2 comments:
The study is interesting for the content and the results are clearly presented. Minor spell check is required in the methods section.
Response
We would like to thank you for reviewing our manuscript and for your helpful suggestions. The changes to the manuscript are shown in red font. We have done a spellcheck and the manuscript has been checked by an English editor.
Reviewer 3 Report
According to the authors, the novelty and potential interest for Antibiotics readers is that this is the first field study in Japan regarding the evaluation of antibiotic prescription patterns using a new indicator DPM.
The statistical analysis provided also sustains reported conclusions.
I have made a few comments, aiming to improve the manuscript quality and readability.
- It is suggested to provide information on those who conducted the study and analysed the data collected (e.g. pharmacists? physicians?, etc..)
- In the Abstract (Line 33) has been mentioned that the received prescriptions were primarily from hospitals or clinics (inpatients). Does it mean 100%? There were no prescriptions from general practices (outpatients)? This is not clear in entirely even in the Methods.
- Line 184 do you need the word used after AMU?
- Line 194-196 Do you have national guidelines describing the use of macrolides in chronic sinusitis as anti-inflammator, and quinolones for otitis media? Or it is based only on Summary of Product Characteristics? Can you clarify this in the text?
- In the line 253 you are writing that the p value over 0.05 indicates statistical significance. Based on this and the tables 2. and 4. in the lines 143-145 “dermatology departments were associated with the use of…” I think it should be “significantly associated” Please clarify the significance.
- Do we generalize this picture with the whole country's perspective?
- Conclusion should relevant to the objectives of the study (line 51-53). Can you highlight these antibiotic classes also in the conclusions? Since, as described in lines 102-107, we can speak about a significant decrease.
Author Response
Reviewer 3 comments:
According to the authors, the novelty and potential interest for Antibiotics readers is that this is the first field study in Japan regarding the evaluation of antibiotic prescription patterns using a new indicator DPM. The statistical analysis provided also sustains reported conclusions. I have made a few comments, aiming to improve the manuscript quality and readability.
Response
We would like to thank you for reviewing our manuscript and for your helpful suggestions. The changes to the manuscript are shown in red font. We have provided a point-by-point response to your comments below.
It is suggested to provide information on those who conducted the study and analyzed the data collected (e.g. pharmacists? physicians? etc..)
Response
According to your suggestion, we have added this information to the Methods section (Page 3, Lines 115-118).
In the Abstract (Line 33) has been mentioned that the received prescriptions were primarily from hospitals or clinics (inpatients). Does it mean 100%? There were no prescriptions from general practices (outpatients)? This is not clear in entirely even in the Methods.
Response
Based on your comments and those of other reviewers, we have added a note on the characteristics of Japanese healthcare and pharmacies in the introduction. (Page 2, Lines 63-74).
Line 184 do you need the word used after AMU?
Response
According to your suggestion, we have deleted the word (Page 10, Line 260).
Line 194-196 Do you have national guidelines describing the use of macrolides in chronic sinusitis as anti-inflammatory, and quinolones for otitis media? Or it is based only on Summary of Product Characteristics? Can you clarify this in the text?
Response
In the Japanese treatment guidelines, macrolides and quinolones are recommended in patients with moderate or severe chronic sinusutus. According to your suggestion, we have revised the text (Page 11, Lines 271-272).
In the line 253 you are writing that the p value over 0.05 indicates statistical significance. Based on this and the tables 2. and 4. in the lines 143-145 “dermatology departments were associated with the use of…” I think it should be “significantly associated” Please clarify the significance.
Response
We have changed the wording as you suggested (Page 5, Lines 188 and 190). We have also revised the Discussion (Page 11, Lines 276-280).
Do we generalize this picture with the whole country's perspective?
Response
Although this study did not collect information from all pharmacies in Japan or a representative sample of pharmacies, it is the largest survey of its kind, and is consistent with the overall national trend, so we believe that the results are broadly generalizable to the whole country.
Conclusion should be relevant to the objectives of the study (line 51-53). Can you highlight these antibiotic classes also in the conclusions? Since, as described in lines 102-107, we can speak about a significant decrease.
Response
Thank you for your constructive suggestion. We have considered your suggestion and the suggestions of the other reviewers and have revised the manuscript to clarify the objectives and conclusions (Page 3, Lines 103-106; Page 11, Line 313 to Page 12, Line 318; Page 12, Lines 320-322).
Reviewer 4 Report
Dear Authors
I'm glad to have the opportunity to review the manuscript. It raises an important topic: assessing the amount of antibiotic use.
The presented data are very interesting, but the manuscript needs improvement. In its current form, it is not sufficient to publish by Antibiotics.
The introduction does not explain the structure of pharmacies in your country. Please indicate the rule of antibiotics dispensing (prescription, OTC, or other). It is incomprehensible to me as a person from abroad. Please describe how pharmacies operate in Japan. e.g., the term "insurance pharmacies" is unknown to me. More context is needed here. Please compare this date with your region and also put more actual references regarding antibiotic use and the role of pharmacies.
The results present exciting data; however, the content lacks references to the new indicators, mentioned by Authors. Please expand this section to emphasize the importance of the proposed indicators. Why did you develop them and how you can use them later?
The discussion sections also lack a more robust grounding of the proposed indicators. Please expand.
In conclusion, it is worth pointing out how the proposed indicators may influence the planning of antibiotic policy in the future. Please indicate the social aspect of the research.
In its present shape, there is no reference to the assumed goal.
Author Response
Reviewer 4 comments:
I'm glad to have the opportunity to review the manuscript. It raises an important topic: assessing the amount of antibiotic use. The presented data are very interesting, but the manuscript needs improvement. In its current form, it is not sufficient to publish by Antibiotics.
Response
We would like to thank you for reviewing our manuscript and for your helpful suggestions. The changes to the manuscript are shown in red font. We have provided a point-by-point response to your comments below.
The introduction does not explain the structure of pharmacies in your country. Please indicate the rule of antibiotics dispensing (prescription, OTC, or other). It is incomprehensible to me as a person from abroad. Please describe how pharmacies operate in Japan. e.g., the term "insurance pharmacies" is unknown to me. More context is needed here. Please compare this date with your region and also put more actual references regarding antibiotic use and the role of pharmacies.
Response
Thank you for pointing this out. In accordance with your comment, we have added some details on the pharmacy system in Japan to the Introduction section (Page 2, Lines 63-67, and Lines 69-74).
The results present exciting data; however, the content lacks references to the new indicators, mentioned by Authors. Please expand this section to emphasize the importance of the proposed indicators. Why did you develop them and how you can use them later?
Response
According to your suggestion, we have revised the Conclusion section to highlight the importance of the indicator (Page 11, Line 313 to Page 12, Line 318)
The discussion sections also lack a more robust grounding of the proposed indicators. Please expand.
In conclusion, it is worth pointing out how the proposed indicators may influence the planning of antibiotic policy in the future. Please indicate the social aspect of the research. In its present shape, there is no reference to the assumed goal.
Response
According to your suggestion and the suggestions of the other reviewers, we have revised the Discussion (Page 10, Lines 225-235) and Conclusion (Page 11, 313 to Page 12, Line 318).
Reviewer 5 Report
Abstract: Consider avoiding the repetition of “according to” in this sentence: Antimicrobial use differed according to the medical department, 40 suggesting the need to target interventions according to the department type. For example, “differed by medical department”
Introduction is followed by Results. Why is the Methods section placed in the end of the manuscript?
Table 1. Line 2: Complete “Number of medical facilities from which the prescriptions …..”
The authors might consider another term for describing the type of department/medical facility as “The detail of specific medical facility” in Table 1.
The following sentences in Discussion are not clear, please rephrase to improve clarity: “Therefore, unlike other settings, the use of narrow-spectrum (targeted) antimicrobials may be promoted in general hospitals. However, since the trend is not observed elsewhere, such as in general hospitals, further 209 support for appropriate use by pharmacy pharmacists is needed. “
The following sentence of Conclusions might be better suited to Discussion: “The decrease in AMU is likely to be due to the combined impact of the National Action Plan, and COVID-19 on AMR control in Japan.
Author Response
Reviewer 5 comments:
Abstract: Consider avoiding the repetition of “according to” in this sentence: Antimicrobial use differed according to the medical department, 40 suggesting the need to target interventions according to the department type. For example, “differed by medical department”
Response
We would like to thank you for reviewing our manuscript and for your helpful suggestions. The changes are shown in red font. We changed the text, as you suggested (Page 1, Line 43).
Introduction is followed by Results. Why is the Methods section placed in the end of the manuscript?
Response
We placed the methods section at the end because we followed the journaltemplate. We checking with the editorial office, and were informed that it was acceptable to place the methods section after the introduction, so we have moved it.
Table 1. Line 2: Complete “Number of medical facilities from which the prescriptions …..”
The authors might consider another term for describing the type of department/medical facility as “The detail of specific medical facility” in Table 1.
Response
We changed the text to “Number of medical facilities/ departments from which the prescriptions were received, per pharmacy per month” based on the suggestion of the English editor (Page 4, Table 1).
The following sentences in Discussion are not clear, please rephrase to improve clarity: “Therefore, unlike other settings, the use of narrow-spectrum (targeted) antimicrobials may be promoted in general hospitals. However, since the trend is not observed elsewhere, such as in general hospitals, further support for appropriate use by pharmacy pharmacists is needed. “
Response
According your suggestion, we have rephrased this section of text to improve the clarity (Page 11, Lines 287-291).
The following sentence of Conclusions might be better suited to Discussion: “The decrease in AMU is likely to be due to the combined impact of the National Action Plan, and COVID-19 on AMR control in Japan.
Response
Thank you for the suggestion. We have considered revised the Discussion and the Conclusions (Page 10, Lines 236-241, and Page 11, Line 313 to Page 12, Line 318, and Page 12, Lines 320-322).
Reviewer 6 Report
The manuscript consists of total 13 pages, 5 tables, 2 figures and the list of total 28 scientific literature references. The article presents original-material based study results concerning the use of antibiotics in Japan based on the analysis of information on dispensing of these drugs provided by pharmacies. As such, the manuscript fits into the spectrum of articles published in the Journal. The title of the manuscript is relevant to the contents of the manuscript. The English language quality is high. The article does not fully conform to the widely accepted structure of a scientific paper as the Authors decided to provide the Readers with a separate Material and methods section but located in an untypical way not before the Results section but only after the Discussion section - the Authors shall consider putting the section in its classical position in order to make the Readers familiar with the clearly defined information on the methods they have chosen, how they have chosen the participating pharmacies and what were the characteristics of their survey; the current organization of the text may improve the reading experience but does not help in maintaining the clarity of the line of argumentation in text.
The Abstract section mirrors the structure and major contents of the main text of the article adequately.
The Introduction provides enough background information to make the context of the study understandable.
The Results section presents in enough detail the findings of the Authors' study that are consistent with the methodology described later; it is significantly aided by the tables and figures.
The Table 1 might gain in clarity in the Authors add to the category description "The detail of specific medical facility, n" the word "...specific prescribing medical...".
The Discussion section is based on the presented study results. What I am missing a bit here is placing the Authors' study results in a bit broader context of the antibiotics consumption also in other countries of the world, which have various and in many cases contradicting experience in this field. The Authors mention the limiting impact on the prescribing practices of the COVID-19 (line 173)and the drop in the number of healthcare ambulatory visits while in some countries the effects were just opposite because of the tele-medicine use resulting in overcautious physicians' prescribing practices while facing the limited diagnostics available this way. The same but because of different reasons happened in some countries at hospitals during the COVID-19 pandemic as physicians tended to apply almost by default "prophylactic antibiotic treatments" to hospitalized COVID-19 patients. It seems that Japan may be among the precious exceptions from the above-mentioned general rules.
The Conclusions section is too general and in my opinion it shall refer to the summed-up specifics of the Authors' results and their practical meaning.
The Authors may consider relating in their article to antimicrobial agents' use patterns and their changes in chosen other then Japan countries during the COVID-19 pandemic, as in e.c. https://doi.org/10.3390/antibiotics11050586 https://doi.org/10.3390/antibiotics11040535 https://doi.org/10.3390/antibiotics11040457 https://doi.org/10.3390/ijerph19074005 https://doi.org/10.3390/antibiotics11040423 https://doi.org/10.3390/medicina58030363 https://doi.org/10.3390/antibiotics11030309 https://doi.org/10.3390/antibiotics11020244 https://doi.org/10.3390/antibiotics10121531 https://doi.org/10.3390/antibiotics10121521 https://doi.org/10.3390/antibiotics10121488
Author Response
Reviewer 6 comments:
The manuscript consists of total 13 pages, 5 tables, 2 figures and the list of total 28 scientific literature references. The article presents original-material based study results concerning the use of antibiotics in Japan based on the analysis of information on dispensing of these drugs provided by pharmacies. As such, the manuscript fits into the spectrum of articles published in the Journal. The title of the manuscript is relevant to the contents of the manuscript. The English language quality is high. The article does not fully conform to the widely accepted structure of a scientific paper as the Authors decided to provide the Readers with a separate Material and methods section but located in an untypical way not before the Results section but only after the Discussion section - the Authors shall consider putting the section in its classical position in order to make the Readers familiar with the clearly defined information on the methods they have chosen, how they have chosen the participating pharmacies and what were the characteristics of their survey; the current organization of the text may improve the reading experience but does not help in maintaining the clarity of the line of argumentation in text.
Response
We would like to thank you for reviewing our manuscript and for your helpful suggestions. The changes are shown in red font. We have provided a point-by-point response to each of your comments below.
In the original manuscript, we place the Material and methods section at the end because we followed the template of the journal Antibiotics. We too prefer to have the Material and methods section in its traditional place, directly after the introduction. After checking with the editorial office, we received a response that it was acceptable to change the placement of the Material and methods section, so we have moved it and inserted it directly after the Introduction. (Page 3, Line 102 ïƒ )
The Abstract section mirrors the structure and major contents of the main text of the article adequately. The Introduction provides enough background information to make the context of the study understandable. The Results section presents in enough detail the findings of the Authors' study that are consistent with the methodology described later; it is significantly aided by the tables and figures. The Table 1 might gain in clarity in the Authors add to the category description "The detail of specific medical facility, n" the word "...specific prescribing medical...".
Response
According your suggestion, we changed the wording in the table (Page 4, Table 1).
The Discussion section is based on the presented study results. What I am missing a bit here is placing the Authors' study results in a bit broader context of the antibiotics consumption also in other countries of the world, which have various and in many cases contradicting experience in this field. The Authors mention the limiting impact on the prescribing practices of the COVID-19 (line 173) and the drop in the number of healthcare ambulatory visits while in some countries the effects were just opposite because of the tele-medicine use resulting in overcautious physicians' prescribing practices while facing the limited diagnostics available this way. The same but because of different reasons happened in some countries at hospitals during the COVID-19 pandemic as physicians tended to apply almost by default "prophylactic antibiotic treatments" to hospitalized COVID-19 patients. It seems that Japan may be among the precious exceptions from the above-mentioned general rules.
Response
Thank you for your advice. According to your suggestion, we have revised Discussion (Page 10, Lines 232-253).
The Conclusions section is too general and in my opinion it shall refer to the summed-up specifics of the Authors' results and their practical meaning.
Response
Thank you for your constructive suggestion. We have revised the Conclusions section based on your input, and that of the other reviewers (Page 11 , Line 313 to Page 12, Line 318; Page 12, Lines 320-322).
The Authors may consider relating in their article to antimicrobial agents' use patterns and their changes in chosen other then Japan countries during the COVID-19 pandemic, as in e.c.
https://doi.org/10.3390/antibiotics11050586, https://doi.org/10.3390/antibiotics11040535
https://doi.org/10.3390/antibiotics11040457, https://doi.org/10.3390/ijerph19074005
https://doi.org/10.3390/antibiotics11040423, https://doi.org/10.3390/medicina58030363
https://doi.org/10.3390/antibiotics11030309, https://doi.org/10.3390/antibiotics11020244
https://doi.org/10.3390/antibiotics10121531, https://doi.org/10.3390/antibiotics10121521
https://doi.org/10.3390/antibiotics10121488
Response
Thank you forsuggesting these references. We have cited these most of them in the revised manuscript.
The details are as follows:
|
Reference |
Reference nr. |
Line nrs. |
|
Rachina, S.; Kozlov, R.; Kurkova, A.; Portnyagina, U.; Palyutin, S.; Khokhlov, A.; Reshetko, O.; Zhuravleva, M.; Palagin, I.; on behalf of Russian Working Group of the Project. Antimicrobial Dispensing Practice in Community Pharmacies in Russia during the COVID-19 Pandemic. Antibiotics 2022, 11, 586. https://doi.org/10.3390/antibiotics11050586 |
15 |
87 385-387 |
|
Jeon, K.; Jeong, S.; Lee, N.; Park, M.-J.; Song, W.; Kim, H.-S.; Kim, H.S.; Kim, J.-S. Impact of COVID-19 on Antimicrobial Consumption and Spread of Multidrug-Resistance in Bacterial Infections. Antibiotics 2022, 11, 535. https://doi.org/10.3390/antibiotics11040535 |
Not cited |
— |
|
Barbieri, E.; Liberati, C.; Cantarutti, A.; Di Chiara, C.; Lupattelli, A.; Sharland, M.; Giaquinto, C.; Hsia, Y.; Doná, D. Antibiotic Prescription Patterns in the Paediatric Primary Care Setting before and after the COVID-19 Pandemic in Italy: An Analysis Using the AWaRe Metrics. Antibiotics 2022, 11, 457. https://doi.org/10.3390/antibiotics11040457 |
23 |
245 409-411 |
|
Kamara, I.F.; Kumar, A.M.V.; Maruta, A.; Fofanah, B.D.; Njuguna, C.K.; Shongwe, S.; Moses, F.; Tengbe, S.M.; Kanu, J.S.; Lakoh, S.; Mansaray, A.H.D.; Selvaraj, K.; Khogali, M.; Zachariah, R. Antibiotic Use in Suspected and Confirmed COVID-19 Patients Admitted to Health Facilities in Sierra Leone in 2020–2021: Practice Does Not Follow Policy. Int. J. Environ. Res. Public Health 2022, 19, 4005. https://doi.org/10.3390/ijerph19074005 |
14 |
87, 247 381-384 |
|
Zayet, S.; Klopfenstein, T. Antibiotics and Therapeutic Agent Prescription in COVID-19 Management. Antibiotics 2022, 11, 423. https://doi.org/10.3390/antibiotics11040423 |
24 |
247 412 |
|
Ferrara, P.; Albano, L. Azithromycin Has Been Flying Off the Shelves: The Italian Lesson Learnt from Improper Use of Antibiotics against COVID-19. Medicina 2022, 58, 363. https://doi.org/10.3390/medicina58030363 |
Not cited |
— |
|
Hek, K.; Ramerman, L.; Weesie, Y.M.; Lambooij, A.C.; Lambert, M.; Heins, M.J.; Hendriksen, J.M.T.; Verheij, R.A.; Cals, J.W.L.; van Dijk, L. Antibiotic Prescribing in Dutch Daytime and Out-of-Hours General Practice during the COVID-19 Pandemic: A Retrospective Database Study. Antibiotics 2022, 11, 309. https://doi.org/10.3390/antibiotics11030309 |
12 |
87 375-377 |
|
Chitungo, I.; Dzinamarira, T.; Nyazika, T.K.; Herrera, H.; Musuka, G.; Murewanhema, G. Inappropriate Antibiotic Use in Zimbabwe in the COVID-19 Era: A Perfect Recipe for Antimicrobial Resistance. Antibiotics 2022, 11, 244. https://doi.org/10.3390/antibiotics11020244 |
25 |
247 413-414 |
|
Borek, A.J.; Maitland, K.; McLeod, M.; Campbell, A.; Hayhoe, B.; Butler, C.C.; Morrell, L.; Roope, L.S.J.; Holmes, A.; Walker, A.S.; Tonkin-Crine, S.; on behalf of the STEP-UP Study Team. Impact of the COVID-19 Pandemic on Community Antibiotic Prescribing and Stewardship: A Qualitative Interview Study with General Practitioners in England. Antibiotics 2021, 10, 1531. https://doi.org/10.3390/antibiotics10121531 |
Not cited |
— |
|
Boeijen, J.A.; van der Velden, A.W.; Hullegie, S.; Platteel, T.N.; Zwart, D.L.M.; Damoiseaux, R.A.M.J.; Venekamp, R.P.; van de Pol, A.C. Common Infections and Antibiotic Prescribing during the First Year of the COVID-19 Pandemic: A Primary Care-Based Observational Cohort Study. Antibiotics 2021, 10, 1521. https://doi.org/10.3390/antibiotics10121521 |
22 |
242 406-408 |
|
Colliers, A.; De Man, J.; Adriaenssens, N.; Verhoeven, V.; Anthierens, S.; De Loof, H.; Philips, H.; Coenen, S.; Morreel, S. Antibiotic Prescribing Trends in Belgian Out-of-Hours Primary Care during the COVID-19 Pandemic: Observational Study Using Routinely Collected Health Data. Antibiotics 2021, 10, 1488. https://doi.org/10.3390/antibiotics10121488 |
13 |
87, 245 378-380 |
Round 2
Reviewer 4 Report
Dear Authors, I’m glad to have the opportunity to review your manuscript. The revisited version contains all needed corrections and in my opinion, is appropriate to publish by Antibiotics